# KIFC1 Inhibitor CW069 Induces Apoptosis and Reverses Resistance to Docetaxel in Prostate Cancer

**DOI:** 10.3390/jcm8020225

**Published:** 2019-02-09

**Authors:** Yohei Sekino, Naohide Oue, Yuki Koike, Yoshinori Shigematsu, Naoya Sakamoto, Kazuhiro Sentani, Jun Teishima, Masaki Shiota, Akio Matsubara, Wataru Yasui

**Affiliations:** 1Department of Molecular Pathology, Graduate School of Biomedical and Health Sciences, Hiroshima University, Hiroshima 734-8551, Japan; akikosekino@gmail.com (Y.S.); b145456@hiroshima-u.ac.jp (Y.K.); yoshis413@yahoo.co.jp (Y.S.); nasakamoto@hiroshima-u.ac.jp (N.S.); kzsentani@hiroshima-u.ac.jp (K.S.); wyasui@hiroshima-u.ac.jp (W.Y.); 2Department of Urology, Graduate School of Biomedical and Health Sciences, Hiroshima University, Hiroshima 734-8551, Japan; teishima@hiroshima-u.ac.jp (J.T.); matsua@hiroshima-u.ac.jp (A.M.); 3Department of Urology, Graduate School of Medical Sciences, Kyushu University, Fukuoka 812-8582, Japan; shiota@uro.med.kyushu-u.ac.jp

**Keywords:** KIFC1, prostate cancer, docetaxel resistance, apoptosis, CW069

## Abstract

Kinesin family member C1 (KIFC1) is a minus end-directed motor protein that plays an essential role in centrosome clustering. Previously, we reported that KIFC1 is involved in cancer progression in prostate cancer (PCa). We designed this study to assess the involvement of KIFC1 in docetaxel (DTX) resistance in PCa and examined the effect of KIFC1 on DTX resistance. We also analyzed the possible role of a KIFC1 inhibitor (CW069) in PCa. We used DTX-resistant PCa cell lines in DU145 and C4-2 cells to analyze the effect of KIFC1 on DTX resistance in PCa. Western blotting showed that KIFC1 expression was higher in the DTX-resistant cell lines than in the parental cell lines. Downregulation of KIFC1 re-sensitized the DTX-resistant cell lines to DTX treatment. CW069 treatment suppressed cell viability in both parental and DTX-resistant cell lines. DTX alone had little effect on cell viability in the DTX-resistant cells. However, the combination of DTX and CW069 significantly reduced cell viability in the DTX-resistant cells, indicating that CW069 re-sensitized the DTX-resistant cell lines to DTX treatment. These results suggest that a combination of CW069 and DTX could be a potential strategy to overcome DTX resistance.

## 1. Introduction

Prostate cancer (PCa) is the most prevalent cancer among men and the second leading cause of cancer-related death in developed countries [1]. Androgen deprivation therapy is initially effective for advanced PCa. However, most of these patients eventually progress to castration-resistant PCa (CRPC), which is a life-threatening disease [2,3]. Docetaxel (DTX) is the standard chemotherapy for CRPC [4]. However, nearly all patients who are treated with DTX become refractory. Therefore, clarifying new molecular mechanisms underlying DTX resistance is necessary to overcome DTX resistance in CRPC.

Increased centrosome number, called centrosome amplification (CA), is a hallmark of human cancer [5]. Recent reports have shown that CA correlates with aneuploidy and malignant behavior in some human cancers including uterine cervical cancer, breast cancer, and PCa [6,7]. Although aneuploidy might cause multipolar spindles and lead to apoptosis, cancer cells overcome these lethal effects of CA by using centrosome clustering. Centrosome clustering, defined as the reshaping of transient multipolar spindles into pseudo-bipolar structures, is a well-studied mechanism that allows cancer cells to avoid apoptosis [8,9]. Kinesin family member C1 (KIFC1) is a minus end-directed motor protein that plays an essential role in centrosome clustering [10,11,12]. Several reports show that KIFC1 is upregulated and is involved in cancer progression in some cancers [13,14,15]. In addition, the overexpression of KIFC1 suppresses DTX-mediated apoptosis in breast cancer cells [16]. Previously, we showed that KIFC1 was associated with a poor prognosis after radical prostatectomy or after DTX treatment in PCa [17]. Additionally, knockdown of KIFC1 improved DTX sensitivity in LNCaP cells and DU145 cells. However, the role of KIFC1 in DTX resistance in PCa is not well known. In this study, we used DTX-resistant PCa cells from C4-2 cells and DU145 cells to analyze the involvement of KIFC1 in DTX resistance. We examined the expression and functional role of KIFC1 and analyzed the effect of KIFC1 knockdown on DTX resistance in DTX-resistant PCa cell lines. We also investigated the effect of the KIFC1 inhibitor CW069 in PCa cell lines.

## 2. Materials and Methods

### 2.1. Cell Lines

Two PCa cell lines (DU145 and C4-2) and DTX-resistant DU145 cells (DU145-DR) and DTX-resistant C4-2 cells (C4-2-DR) were kindly provided by Dr. Masaki Shiota (Kyushu University, Fukuoka, Japan). The DU145 cell lines were maintained in MEM (Nissui Pharmaceutical Co. Ltd., Tokyo, Japan), and the C4-2 cells were maintained in RPMI 1640 (Nissui Pharmaceutical Co. Ltd.) containing 10% fetal bovine serum (BioWhittaker, Walkersville, MD, USA), 2 mM L-glutamine, 50 U/mL penicillin, and 50 g/mL streptomycin in a humidified atmosphere of 5% CO_2_ at 37 °C. DTX cell lines were cultured under DTX at a dose of 2 ng/mL for DU145-DR and 5 ng/mL for C4-2-DR.

### 2.2. DTX and CW069 Treatment

DTX was obtained from Sanofi-Aventis and handled according to the manufacturer’s recommendations [18]. CW069 was obtained from Funakoshi (Tokyo, Japan). Twenty-four hours after transfection of siRNAs for KIFC1 or negative control, these cells were exposed to DTX for 48 hr. Cell viability was measured by an MTT assay. An MTT assay was performed 48 h after DTX or CW069 treatment. Drug sensitivity curves and IC50 values were calculated using GraphPad Prism 4.0 software (GraphPad Software) [17].

### 2.3. Western Blotting Analysis

For Western blotting analysis, cells were lysed as described previously [19]. Primary antibody, KIFC1 (H00003833-M01, Abnova, Taipei, Taiwan), Bcl-2 (sc-7382, Santa Cruz Biotechnology, Santa Cruz, CA, USA), Bax (sc-7480, Santa Cruz Biotechnology, Santa Cruz, CA, USA), cleaved PARP (c-PARP) (#5625, Cell Signaling Technology, Inc., Danvers, MA, USA), cleaved caspase-3 (c-caspase-3) (#9661, Cell Signaling Technology, Inc., Danvers, MA, USA) were used. β-Actin (Sigma-Aldrich, St. Louis, MO, USA) was used as a loading control.

### 2.4. qRT-PCR Analysis

Total RNA was isolated from frozen cancer cell lines using Isogen (Nippon Gene, Tokyo, Japan), and 1 μg of total RNA was converted to cDNA with a first-strand cDNA synthesis kit (Amersham Biosciences Corp., Piscataway, NJ, USA). The qPCR was performed with a SYBR Select Master Mix (Applied Biosystems, Austin, TX, USA) as described previously [20]. ACTB-specific PCR products, which were amplified from the same RNA samples, served as internal controls. KIFC1 primer sequence: forward primer GACGCCCTGCTTCATCTG; reverse primer CCAGGTCCACAAGACTGAGG.

### 2.5. RNA Interference

Silencer^®^ Select (Ambion, Austin, TX, USA) against KIFC1 was used for RNA interference. Two independent oligonucleotides and negative control small interfering RNA (siRNA) (Invitrogen, Carlsbad, CA, USA) were used. Transfection was performed using Lipofectamine RNAiMAX (Invitrogen) according to the manufacturer’s instructions. Cells were used 48 h after transfection in each of the experiments and assays [17].

### 2.6. Cell Death ELISA

Cells were seeded in 12-well plates (1 × 10^5^ cells) and treated as indicated. Mono- and oligonucleosomes in the cytoplasmic fraction were measured by a cell death detection ELISA kit (Roche, Basel, Switzerland) according to the manufacturer’s instructions. Absorbance was determined at 405 nm.

### 2.7. Statistical Analysis

Statistical differences were evaluated using a two-tailed Student *t*-test or Mann-Whitney U-test. A *p*-value of <0.05 was considered statistically significant. Statistical analyses were conducted primarily using GraphPad Prism software (GraphPad Software Inc., La Jolla, CA, USA). The combination index (CI) was calculated by the Chou–Talalay method. A combination index (CI) < 1 indicates synergism, CI = 1 an additive effect, and CI > 1 an antagonistic effect [21].

## 3. Results

### 3.1. Characterization of DTX-Resistant PCa Cell Lines

We used C4-2-DR and DU145-DR cells to analyze the involvement of KIFC1 in DTX resistance. MTT assays were performed to measure cell viability under various concentrations of DTX in the C4-2-DR and DU145-DR cells. The IC50 values of the C4-2-DR and DU145-DR cells were significantly higher than those of the parental DU145 and C4-2 cells, which was consistent with previous results (Figure 1A) [22,23]. We compared the expression of c-PARP, which was used as a marker of apoptosis in the parental and DTX-resistant cell lines. Western blotting showed that the expression of c-PARP and c-caspase-3 was induced by DTX treatment in the parental DU145 and C4-2 cells. On the contrary, the expression of c-PARP was not changed by DTX treatment in DU145-DR and C4-2-DR cells (Figure 1B). These results suggest that the DU145-DR and C4-2-DR cells were resistant to DTX treatment.

### 3.2. KIFC1 is Overexpressed in DTX-Resistant Cell Lines

To verify whether KIFC1 is involved in DTX resistance, we investigated the expression of KIFC1 in DU145-DR and C4-2-DR cells. Western blotting and qRT-PCR showed that KIFC1 was overexpressed in DU145-DR and C4-2-DR cells compared with the parental DU145 and C4-2 cells at both mRNA and protein levels (Figure 2A,B).

### 3.3. Inhibition of KIFC1 Induces Apoptosis Pathway and Reverses DTX Resistance In Vitro

Several studies have shown that KIFC1 is associated with an apoptosis pathway [24,25]. We used RNA interference targeting KIFC1 in DU145-DR and C4-2-DR cells and confirmed the efficiency of KIFC1 knockdown by Western blotting (Figure 3A). Western blotting showed that inhibition of KIFC1 enhanced the expression of Bax2, c-PARP, and c-caspase-3 and reduced the expression of Bcl-2 in DU145-DR and C4-2-DR cells (Figure 3A). Given that KIFC1 was overexpressed in the DTX-resistant cell lines and is involved in the apoptosis pathway, we next analyzed whether the knockdown of KIFC1 improves DTX sensitivity in DU145-DR and C4-2-DR cells. We measured cell viability in DU145-DR and C4-2-DR cells with knockdown of KIFC1 under various concentrations of DTX. We found that downregulation of KIFC1 re-sensitized DU145-DR and C4-2-DR cells to DTX treatment (Figure 3B).

### 3.4. Effect of KIFC1 Inhibitor CW069 on Cell Viability

A recent study reported that CW069 is a novel and allosteric inhibitor of KIFC1 [26]. To clarify the effect of CW069 on cell viability in PCa, we measured cell viability under various concentrations of CW069 in both parental and DTX-resistant cell lines. CW069 treatment suppressed cell viability in both the parental and DTX-resistant cell lines (Figure 4A). The IC50 values of the DTX-resistant cell lines treated with CW069 were significantly lower than those of the parental cell lines, suggesting that the effect of CW069 on cell viability may depend on the expression of KIFC1. Next, to test whether CW069 could selectively suppress cell viability in cancer cells, we investigated the effect of CW069 in RWPE-1 cells, which is a normal prostate epithelial cell line [27]. Western blotting demonstrated that the expression of KIFC1 was not detected in RWPE-1 cells (Figure 4B). As we expected, CW069 treatment had little effect on cell viability in RWPE-1 cells compared with the DU145 and C4-2 cells (Figure 4C). Furthermore, we performed a cell death ELISA assay to analyze the ability of CW069 to induce apoptotic cell death in RWPE-1, DU145, and C4-2 cells. CW069 treatment had little effect on apoptotic cell death in the RWPE-1 cells but had a significant effect on the DU145 and C4-2 cells (Figure 4D).

### 3.5. CW069 Re-Sensitizes DTX-Resistant Cell Lines to DTX Treatment

As shown in Figure 3B, knockdown of KIFC1 reversed DTX resistance. Therefore, we investigated the effect of combination therapy with DTX and CW069. We measured cell viability under DTX alone or in combination with CW069 in parental and DTX-resistant cell lines. DTX alone had little effect on cell viability in the DTX-resistant cell lines. However, the combination of DTX and CW069 significantly reduced cell viability in the DTX-resistant cell lines (Figure 5A). The cell death ELISA assay showed that the combination of DTX and CW069 led to significant induction of apoptosis compared to DTX alone in the DTX-resistant cell lines (Figure 5B). In addition, we analyzed the dose response for the combination of DTX and CW069 in the DTX-resistant cell lines and calculated the combination index to assess whether the combination of DTX and CW069 is synergistic or additive. A synergistic effect was observed in the DTX-resistant cell lines (Table 1).

## 4. Discussion

DTX has been the first-line therapy for metastatic CRPC patients since 2004. Recent clinical studies have reported that early DTX treatment combined with androgen deprivation therapy results in improved overall survival in comparison to androgen deprivation therapy alone in patients with metastatic hormone-sensitive PCa [28,29]. This finding suggests that the beneficial effect of DTX may not be restricted to CRPC and that DTX treatment is becoming increasingly more important in PCa [30]. Although DTX treatment improves overall survival, disease relapse eventually occurs due to the development of DTX resistance [31]. Several factors have been shown to be involved in DTX resistance [30,32]. Loss of p53 leads to DTX resistance, and p53 status is an essential determinant of DTX sensitivity [33]. Recent evidence has shown that alteration of β-tubulin isotypes is correlated with DTX resistance [34]. In addition, the expression of multidrug-resistant proteins such as ABCB1 is upregulated in a DTX-resistant PCa cell line. However, these above molecules have not been utilized clinically. Therefore, there is an urgent need to clarify the mechanisms of DTX resistance. A recent study reported that the expression of KIFC1 is upregulated in DTX-resistant breast cancer cell lines compared with that of DTX-sensitive cell lines. What is more, overexpression of KIFC1 increased the pools of free tubulin and promoted DTX resistance in breast cancer [16]. This evidence suggests that KIFC1 may antagonize the effect of DTX at least through the dissociation of tubulin from microtubules. In the present study, the expression of KIFC1 was upregulated in DTX-resistant PCa cell lines. Knockdown of KIFC1 re-sensitized the DTX-resistant cells to DTX treatment in DU145 and C4-2 cells. To date, some preclinical studies have addressed the finding that anti-apoptotic proteins regain sensitivity to DTX [30]. ABT-263, which is a Bcl-2 inhibitor, restored DTX sensitivity in DTX-resistant cells in PCa [35]. Furthermore, glucocorticoid receptor antagonism also re-sensitizes DTX resistance through a reduction of BcL-xL expression [36]. In the present study, knockdown of KIFC1 suppressed the expression of Bcl-2, cleaved PARP and cleaved caspase-3, and enhanced the expression of Bax. This result indicates a potential mechanistic explanation for the restoration of DTX sensitivity in PCa.

A recent study reported that CW069 was identified as a highly selective small-molecule KIFC1 inhibitor using a chemogenomics-based approach [26]. CW069 increases multipolar spindle formation and inhibits cell viability in cancer cells in breast cancer. However, to date, there have been few reports on CW069 [26,37]. In the present study, CW069 treatment selectively damaged parental and DTX-resistant PCa cells but had little effect on cell viability in RWPE-1 cells. The result that CW069 re-sensitized DTX-resistant cell lines to DTX treatment has potential clinical implications. In addition, the synergistic effect was found in the combination of DTX and CW069. In current cancer treatments, different types of chemotherapeutic agents are combined to improve efficacy and to minimize toxicity. Our previous study showed that the expression of KIFC1 was higher in PCa tissues than in various normal tissue samples [17]. Collectively, these results suggest that a combination of DTX and CW069 may be a promising therapy for CRPC patients that causes fewer adverse effects.

There are some limitations in this study. First, so far, three KIFC1 inhibitors (CW069, AZ82, and SR31527) have been reported [26,38,39]. Although these three drugs have been shown to lead to multipolar mitosis and decrease cell viability in human cancer, their effects were somewhat different because each drug binds to a different allosteric site on KIFC1 [38]. Furthermore, using a unique assay, a recent study showed that these drugs might not be specific to KIFC1 [40] Therefore, further study using these three drugs in PCa will be necessary in the future to verify our current findings. Second, recent studies have shown that cross-resistance of DTX cells were resistant to both DTX and cabazitaxel [41,42]. However, in this study, we focused on the role of KIFC1 and KIFC1 inhibitor (CW069) on only DTX resistance in PCa. In the near future, we will investigate the role of KIFC1 on cross-taxan resistance in PCa.

## 5. Conclusions

In conclusion, we used DTX-resistant PCa cell lines to analyze the role of KIFC1 in DTX resistance. We found that the expression of KIFC1 was significantly upregulated in cells with DTX resistance. Inhibition of KIFC1 induced an apoptosis pathway and re-sensitized the cellular response to DTX. Additionally, CW069 re-sensitized DTX-resistant cells to DTX treatment. The data presented here emphasize the great potential of combination therapy with DTX and CW069 in the treatment of PCa.

## Figures and Tables

**Figure 1 jcm-08-00225-f001:**
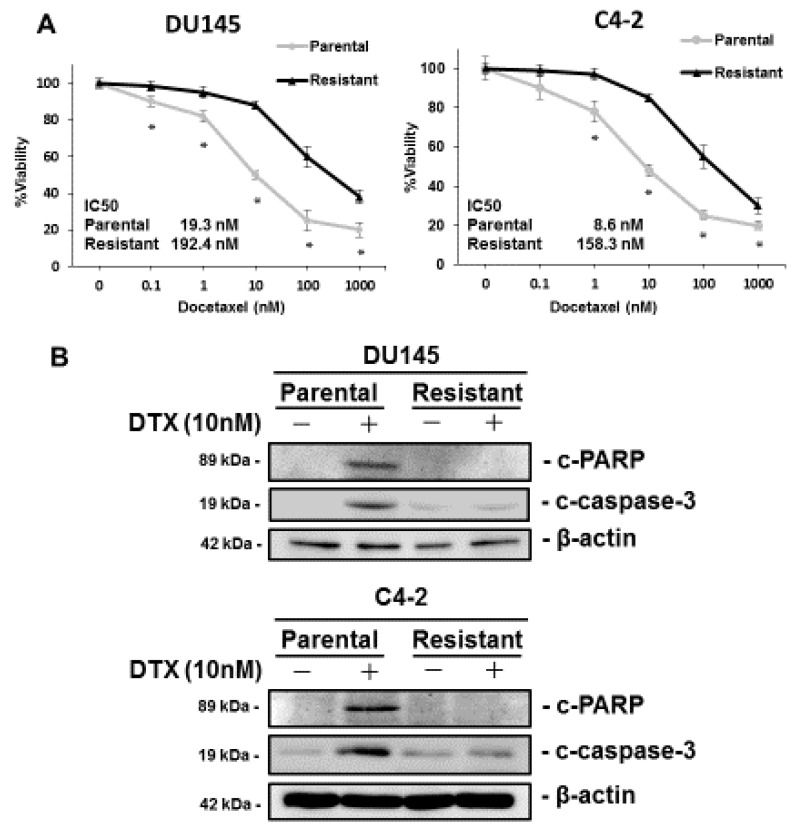
Characterization of docetaxel (DTX)-resistant prostate cancer cell lines. (**A**) The dose-dependent effects of DTX on the viability of parental and DTX-resistant cell lines in DU145 and C4-2 cells. The results are expressed as the mean and S.D. of triplicate measurements. * *p* < 0.01. (**B**) Western blotting of c-PARP and c-caspase-3 in parental and DTX-resistant cell lines in DU145 and C4-2 cells in the presence of DTX (10 nM) or vehicle (ethanol). β-actin was used as a loading control. c-PARP: cleaved PARP; c-caspase-3: cleaved caspase-3.

**Figure 2 jcm-08-00225-f002:**
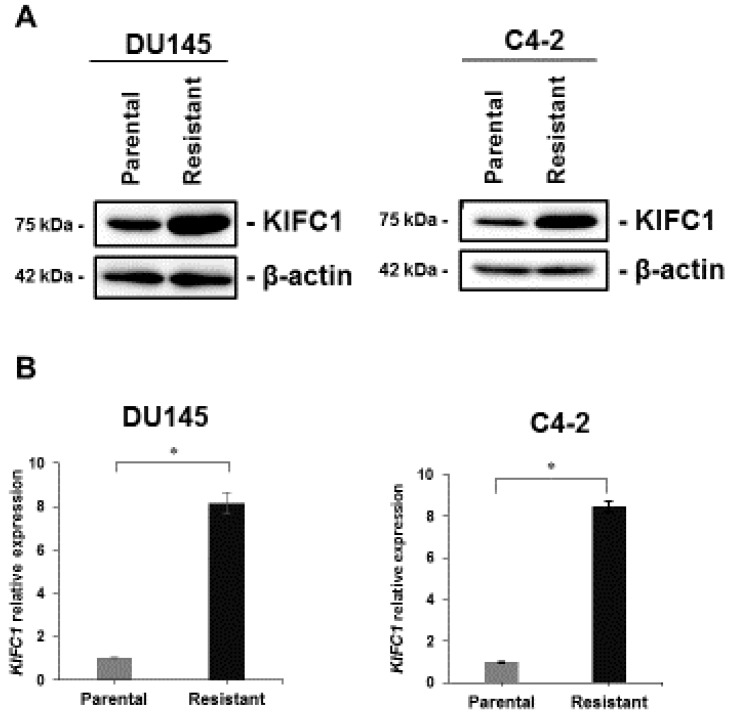
KIFC1 is overexpressed in docetaxel (DTX)-resistant cell lines and in a castration-resistant prostate cancer (CRPC) patient. (**A**) Western blotting of KIFC1 in parental and DTX-resistant cell lines. β-actin was used as a loading control. (**B**) qRT-PCR of KIFC1 in parental and DTX-resistant cell lines. The results are expressed as the mean and S.D. of triplicate measurements. * *p* < 0.01.

**Figure 3 jcm-08-00225-f003:**
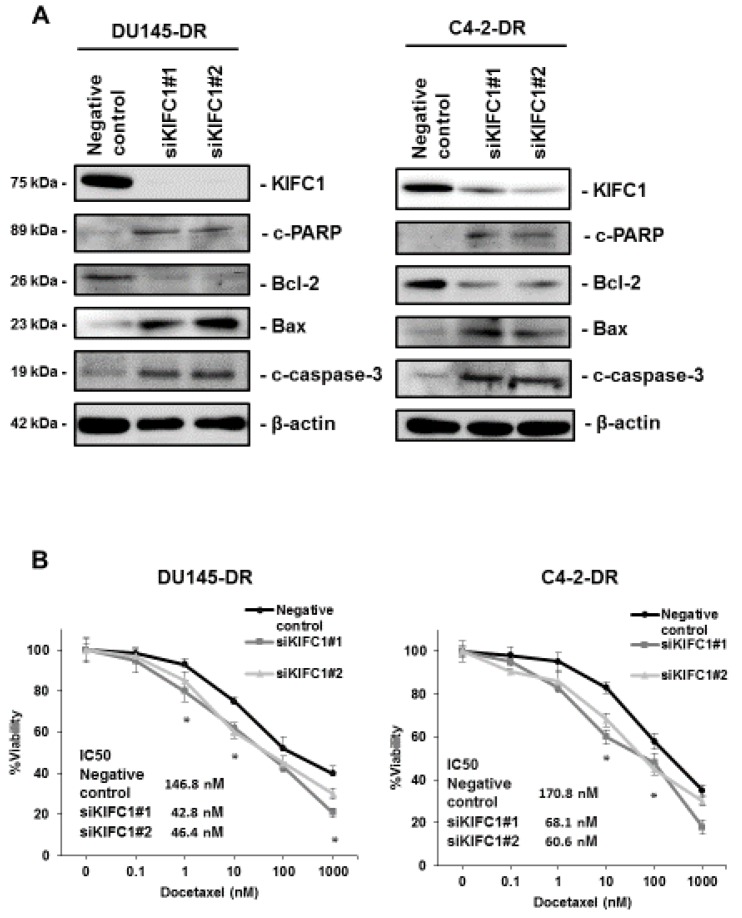
Inhibition of KIFC1 induces an apoptosis pathway and reverses docetaxel (DTX) resistance in vitro. (**A**) Western blotting of KIFC1, c-PARP, Bcl-2, Bax, and c-caspase-3 in DU145-DR and C4-2-DR cells transfected with a negative control or two different siRNAs for KIFC1. β-actin was used as a loading control. c-PARP: cleaved PARP; c-caspase-3: cleaved caspase-3 (**B**) The dose-dependent effects of DTX on the viability of DU145-DR and C4-2-DR cells transfected with negative control or two different siRNAs for KIFC1. The results are expressed as the mean and S.D. of triplicate measurements. * *p* < 0.01.

**Figure 4 jcm-08-00225-f004:**
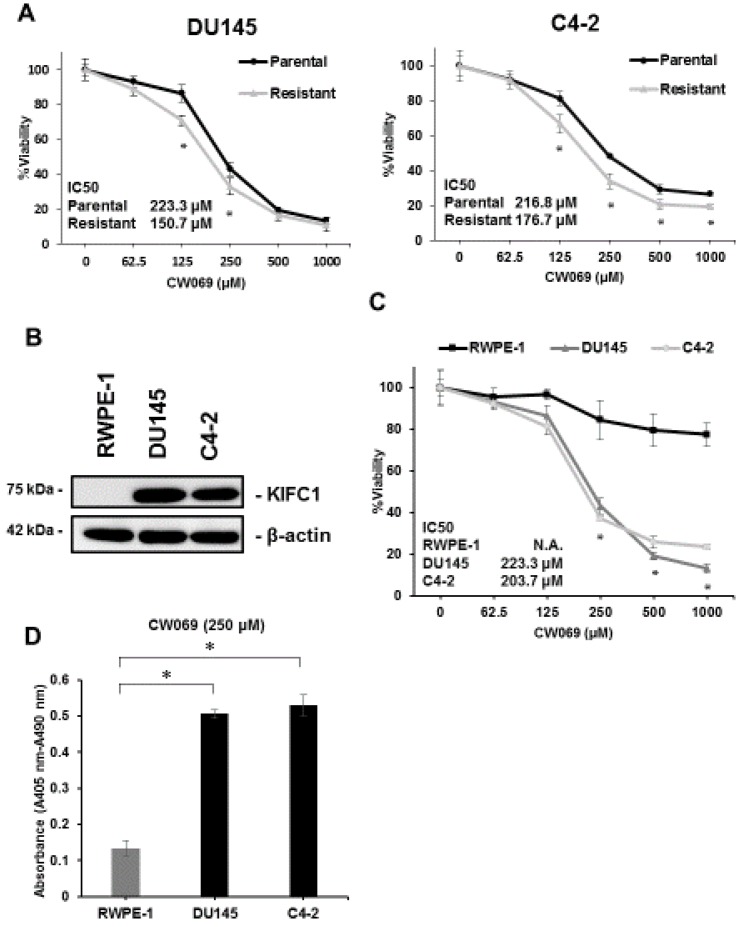
The effect of the KIFC1 inhibitor CW069 on cell viability (**A**) The dose-dependent effects of CW069 on cell viability in parental and docetaxel-resistant cell lines in DU145 and C4-2 cells. The results are expressed as the mean and S.D. of triplicate measurements. * *p* < 0.01. (**B**) Western blotting of KIFC1 in RWPE-1, DU145, and C4-2 cells. (**C**) The dose-dependent effects of CW069 on cell viability in RWPE-1, DU145, and C4-2 cells. The results are expressed as the mean and S.D. of triplicate measurements. * *p* < 0.01. (**D**) The cell death ELISA in RWPE-1, DU145, and C4-2 cells treated with CW069 (250 μM). The results are expressed as the mean and S.D. of triplicate measurements. * *p* < 0.01.

**Figure 5 jcm-08-00225-f005:**
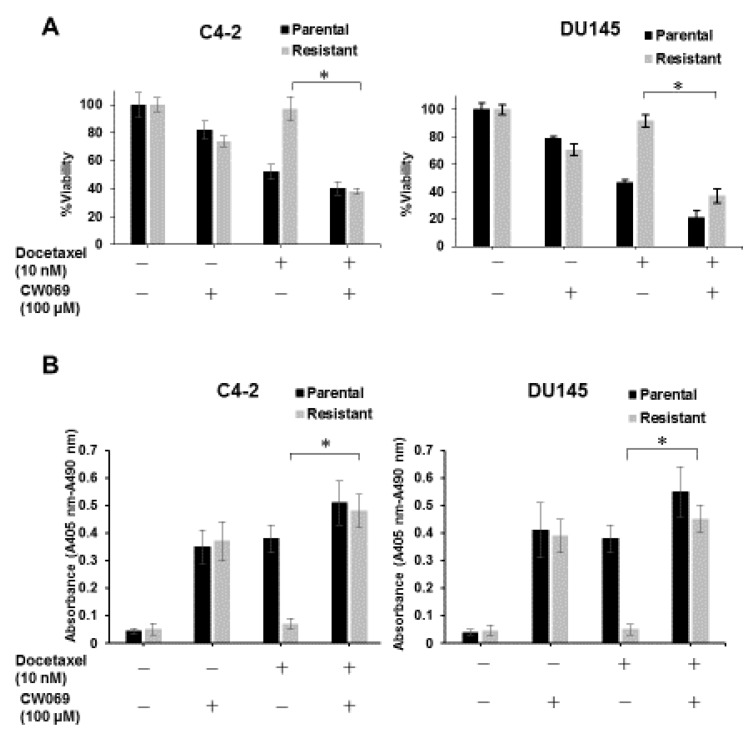
CW069 re-sensitizes DTX-resistant cell lines to docetaxel (DTX) treatment. (**A**) The effect of the combination of DTX and CW069 on cell viability in parental and DTX-resistant cell lines in DU145 and C4-2 cells. The results are expressed as the mean and S.D. of triplicate measurements. * *p* < 0.01. (**B**) The effect of the combination of DTX and CW069 on apoptosis in parental and DTX-resistant cell lines in DU145 and C4-2 cells. The results are expressed as the mean and S.D. of triplicate measurements. * *p* < 0.01.

**Table 1 jcm-08-00225-t001:** The combination index (CI) values for the combination of docetaxel (DTX) and CW069 in DTX-resistant cell lines.

DU145-DR		Docetaxel (nM)		
		5	10	20
CW069 (μM)	50	0.55	0.45	0.45
	100	0.73	0.48	0.54
	200	0.72	0.71	0.47
C4-2-DR		Docetaxel (nM)		
		5	10	20
CW069 (μM)	50	0.66	0.51	0.62
	100	0.77	0.46	0.53
	200	0.66	0.67	0.42

CI: combination index; DU145-DR: docetaxel-resistant DU145; C4-2-DR: docetaxel-resistant C4-2; CI = 1: an additive effect; CI < 1: a synergistic effect.

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
