# Peer review of "KIFC1 Inhibitor CW069 Induces Apoptosis and Reverses Resistance to Docetaxel in Prostate Cancer"

_jcm, 2019, doi:10.3390/jcm8020225_

Reviewer 1 Report

This is a rather simplistic manuscript that seeks to define a role of Kinesin KIFC1 inhibitor in the treatment of two docetaxel resistant cancer cell lines. I find several weaknesses with the study, primarily the lack of innovation and total absence of any mechanistic depth, thus a questionable translational significance, if the goal is to identify this kinesin as a therapeutic target. In addition I have the following issues: 

A few remarks:

There should be statistics shown on Figure 1A, 3B, and 4C. 

While I have concerns of the authors only using one patient sample, I accept that they do not draw conclusions in their text from it and rather present the patient sample as rather anecdotal data.  I assume that if the authors had access to more than one sample, they would have analyzed all of them and presented their data as such. However this still presents a major limitation as to the interpretation of the finding and the clinical relevance of the work.

There is no magnification or scale bar for their immunostaining images shown on Figure 2C.

While c-PARP cleaved analysis is acceptable as a marker of apoptosis, it is still is an incomplete marker of apoptosis and I suggest that the authors performed analysis of activated cleaved caspase to complement their findings since apoptosis is the major endpoint. 

Figure 4D is not described/labeled in the Results section or anywhere throughout the text.

The following sentences in the discussion (Pg  8 of 11, towards the bottom) are particular concerning as they indicate the lack of expertise/knowledge and the ignorance by the authors:

“However, the mechanism of action of DTX has not been fully elucidated” It is well established that docetaxel as a major  first line taxane chemptherapy for prostate cancer works through microtubule stabilization, apoptosis induction, and through blocking AR translocation to the nucleus.  It is inappropriate and unacceptable for the authors to make such claims.  Instead they should consider stating something like “there are many mechanisms of DTX action on prostate cancer cells”

“However, the mechanism of how KIFC1 mediated the DTX resistance could not be fully elucidated” –This is a wrong and miseleading statement. The action of kinesin in mitotic division and centromere amplification CAN be fully elucidated and have been targeted by novel effective chemotherapeutic inhibitors. However not fully elucidated in this paper. Thus failing to deliver as to the title of the manuscript.

Author Response

Our responses to the reviewer's comments for No: jcm-424770

TitleKIFC1 Inhibitor CW069 Induces Apoptosis and Reverses Resistance to Docetaxel in Prostate Cancer

AuthorsYohei Sekino, Naohide Oue, Yuki Koike, Yoshinori Shigematsu, Naoya Sakamoto, Kazuhiro Sentani, Jun Teishima, Masaki Shiota, Akio Matsubara, Wataru Yasui

ReviewersWe thank the reviewers for the critical comments which have helped us to improve the manuscript. We have addressed all criticisms as follows.

Reviewer #1

1. The reviewer required us to show statistics on Figure1A, 3B, and 4C.

We thank the reviewer for the valuable comments. We added statistical significance to Figure1A, 3B, and 4C.

These statements were described in the revised Figure1A, 3B, and 4C.

2. The reviewer proposed that we should use more samples in IHC analysis.

We thank the reviewer for the important suggestion. Because samples obtained before and after docetaxel are very rare resources, we could analyze only one sample in IHC analysis. Unfortunately, we do not have more samples for now. We totally agree that one could not draw any conclusion based on one sample. Therefore, we excluded the result of IHC analysis from this study. In the near future, we will surely perform IHC analysis of KIFC1 in much more samples.

3. The reviewer requested us to analyze cleaved caspase-3 in addition to cleaved PARP by western blotting.

We thank the reviewer for the valuable assignment. We agree that analyzing only cleaved PARP is insufficient for apoptosis analysis. We analyzed the expression of cleaved caspase-3 by western blotting. Knockdown of KIFC1 induced the expression of cleaved caspase-3. the expression of cleaved-caspase-3 was induced by docetaxel treatment in the parental DU145 and C4-2 cells

These statements were described in the revised results (page 3, line 24 and page 5, line 10).

4. The reviewer requested us to modify some statements in discussion.

We thank the reviewer for the valuable comments and agree with the reviewer’s comments. We rewrote the statement “However, the mechanism of action of DTX has not been fully elucidated” and excluded the statement “However, the mechanism of how KIFC1 mediated the DTX resistance could not be fully elucidated”

These statements were described in the revised discussion. (page 8, line 14-15).

Reviewer 2 Report

This manuscript reports that the kinesin protein KIFC1 is upregulated in prostate cancer cells resistant to docetaxel. The data are largely consistent with the authors’ conclusion. However, the significance and potential impact are uncertain due to limited scope and scientific rigor in experimental design. For example, the authors only presented one sample in immunohistochemistry analysis of KIFC1 in a PCa patient tumor (Fig. 2C). One could not draw any conclusion based on one sample. Moreover, the data were drawn from a few cell lines (C4-2, DU145 and RWPE1). Inclusion of more cell lines will improve scientific soundness. The authors are also encouraged to study in vivo effects of KIFC1 on docetaxel treatment response. The depth of experiments should be improved. The authors narrowly focused on KIFC1 without examining other members of the kinesin family. Since KIFC1 mRNA levels were markedly increased in docetaxel resistant cell lines (Fig. 2B), RNA-seq profiling may be informative to understand potential mechanism(s) underlying docetaxel resistance. This may also shed light on how KIFC1 may regulate apoptosis regulators (Fig. 3A).

Minor points

1.     Please include the vendor catalog numbers for all antibodies as well as the primer sequences for qRT-PCR in the materials and methods.

2.     Molecular weights should be included in all immunoblotting images.

3.     The IC50 values are nM, not µM in Fig. 3B.

4.     I just cannot see any difference of the CW069 between docetaxel resistance cells and parental cells (Fig. 4A), which raises a concern regarding the specificity of CW069 against KIFC1 as the authors also discussed in the manuscript.

5.     Dose response for the docetaxel and CW069 combination must be performed (Fig. 5). The authors should also assess whether the combination is synergistic or additive.

Author Response

Our responses to the reviewer's comments for No: jcm-424770

TitleKIFC1 Inhibitor CW069 Induces Apoptosis and Reverses Resistance to Docetaxel in Prostate Cancer

AuthorsYohei Sekino, Naohide Oue, Yuki Koike, Yoshinori Shigematsu, Naoya Sakamoto, Kazuhiro Sentani, Jun Teishima, Masaki Shiota, Akio Matsubara, Wataru Yasui

ReviewersWe thank the reviewers for the critical comments which have helped us to improve the manuscript. We have addressed all criticisms as follows.

Reviewer #2

1. The reviewer required us to show the catalog numbers for all antibodies and primer sequences for qRT-PCR.

We thank the reviewer for the valuable comments. We added the information of antibody and primer sequence to materials and methods.

These statements were described in the revised materials and methods (page 2, line 29-33 and 41-42).

2. The reviewer proposed that we should write molecular weights in immunoblotting images.

We thank the reviewer for the important suggestion. We added the information of molecular weights to immunoblotting images.

3. The reviewer requested us to modify erratum.

We are terribly sorry for these silly mistakes. We have closely reviewed and revised them.

4. The reviewer requested us to analyze the effect of CW069 on docetaxel resistance cells and parental cells again.

We thank the reviewer for the valuable assignment. We investigated the effect of CW069 on docetaxel resistance cells and parental cells again and observed that IC50 was significantly lower in docetaxel resistant cells that parental cells in DU145 and C4-2 cells. This finding suggests that the effect is according to the expression of KIFC1.

These statements were described in the revised result (page 6, line 6).
5. The reviewer requested us to analyze dose response for the docetaxel and CW069 combination to assess whether the combination is synergistic or additive.

We thank the reviewer for the essential suggestion. We performed dose response for the combination of docetaxel and CW069 and observed that the combination of docetaxel and CW069 was synergistic.

These statements were described in the revised results and table 1 (page 7, line 8-11).

Reviewer 3 Report

In the present study, the authors identified the functional role of an up-regulation of the KIFC1 protein  in docetaxel resistance in prostate cancer cells using two established cell lines (Du145 and C4-2). While the story is short and interesting, two main questions remains. Is KIFC1 up-regulation also confers resistance to cabazitaxel? This is an important question as other groups have showed some cross-resistance of DocR cells for both Doc and Cab. While the authors stated that KIFC1 expression level has been linked to centrosome clustering and microtubules network, these two are not evaluated in the study.

Author Response

Our responses to the reviewer's comments for No: jcm-424770

TitleKIFC1 Inhibitor CW069 Induces Apoptosis and Reverses Resistance to Docetaxel in Prostate Cancer

AuthorsYohei Sekino, Naohide Oue, Yuki Koike, Yoshinori Shigematsu, Naoya Sakamoto, Kazuhiro Sentani, Jun Teishima, Masaki Shiota, Akio Matsubara, Wataru Yasui

ReviewersWe thank the reviewers for the critical comments which have helped us to improve the manuscript. We have addressed all criticisms as follows.

Reviewer #3

1. The reviewer required us to analyze the expression of KIFC1 in cabazitaxel resistant cell lines.

We thank the reviewer for the valuable suggestion. We have already confirmed that the expression of KIFC1 was upregulated in cabazitaxel resistant prostate cancer cell lines. However, in this study, we focused on the role of KIFC1 and KIFC1 inhibitor (CW069) on docetaxel resistance in prostate cancer. In the near future, we will report the role of KIFC1 on cross-taxan resistance in prostate cancer.

These statements were described in the revised discussion (page 9, line 2-6).

2. The reviewer proposed that we should analyze the association between KIFC1 and centrosome clustering and microtubules network.

We thank the reviewer for the important comment. As we mentioned in the introduction, several studies have shown that KIFC1 play an essential role in centrosome clustering and microtubule network in some cancer. However, the reference is insufficient to support above evidence. We added some reference to the introduction.

These statements were described in the revised introduction (page 2, line 2).

Reviewer 4 Report

The study by Sekino and co-authors investigates the role of the Kinesin family member C1 (KIFC1), a minus end-directed motor protein playing a pivotal role in centrosome clustering, in docetaxel (DTX) resistance in prostate cancer (PCa) cells. In particular, they found that KIFC1 expression was higher in DTX-resistant cell lines than that in the parental ones, and that downregulation of KIFC1 re-sensitized the DTX-resistant cell lines to DTX treatment. A similar result was observed using the KIFC1 inhibitor, CW069, that suppressed cell viability in both parental and DTX-resistant cell lines. Importantly, the combination of DTX and CW069 significantly reduced cell viability in the DTX-resistant cells, indicating that CW069 re-sensitized the DTX-resistant cell lines to DTX treatment. Altogether these results suggest that a combination of CW069 and DTX could be a potential strategy to overcome DTX resistance.

Overall, the present study is interesting and significantly increases our understanding on the molecular mechanisms involved in PCa progression. More importantly, the results here presented open new alleys of investigation towards the employment of novel potential therapeutic agents in the treatment of advanced PCa. In general, Methods are appropriate, Results are adequately described and discussed also in the context of the literature. Experiments have been appropriately performed to demonstrate the hypotheses and data have been interpreted logically to arrive at conclusions. English language is also appropriate.

Minor comments

1) Introduction. It would be of benefit to add the following recent Reference “J Cell Mol Med. 2018 May;22(5):2865-2883. doi: 10.1111/jcmm.13581” close to Ref.2

2) Methods: they should be described in more details

3) I strongly believe that the IHC analysis performed considering only one patient is very limiting. I would extend the size to at least 10 samples

4) When evaluating apoptosis, I would detect also caspase-3 in addition to c-PARP, Bcl-2 and Bax and provide morphological data (DNA fragmentation, for instance)

Author Response

Our responses to the reviewer's comments for No: jcm-424770

TitleKIFC1 Inhibitor CW069 Induces Apoptosis and Reverses Resistance to Docetaxel in Prostate Cancer

AuthorsYohei Sekino, Naohide Oue, Yuki Koike, Yoshinori Shigematsu, Naoya Sakamoto, Kazuhiro Sentani, Jun Teishima, Masaki Shiota, Akio Matsubara, Wataru Yasui

ReviewersWe thank the reviewers for the critical comments which have helped us to improve the manuscript. We have addressed all criticisms as follows.

Reviewer #4

1. The reviewer required us to add one reference in introduction.

We thank the reviewer for the valuable comments. We added the reference to introduction.

These statements were described in the revised introduction (page1, line 36).

2. The reviewer proposed that we should describe the detail of materials and methods.

We thank the reviewer for the important suggestion. We described materials and methods in more detail.

3. The reviewer proposed that we should use more samples in IHC analysis.

We thank the reviewer for the important suggestion. Because samples obtained before and after docetaxel are very rare resources, we could analyze only one sample in IHC analysis. Unfortunately, we do not have more samples for now. We totally agree that one could not draw any conclusion based on one sample. Therefore, we excluded the result of IHC analysis from this study. In the near future, we will surely perform IHC analysis of KIFC1 in much more samples.

4. The reviewer requested us to analyze cleaved caspase-3 in addition to cleaved PARP by western blotting.

We thank the reviewer for the valuable assignment. We agree that analyzing only cleaved PARP is insufficient for apoptosis analysis. We analyzed the expression of cleaved caspase-3 by western blotting. Knockdown of KIFC1 induced the expression of cleaved caspase-3. We could not perform morphological analysis due to budgetary issue.

These statements were described in the revised results (page 3, line 24 and page 5, line 10).